# The impact of housing prices and land financing on economic growth: Evidence from Chinese 277 cities at the prefecture level and above

Qian Sun[1,2], Sohail Ahmad Javeed[ORCID][3]*, Yong Tang[1,2], Yan Feng[1,2]

**1** Management School, Hunan City University, Yiyang, China, **2** Hunan New Urbanization Research Institute, Yiyang, China, **3** School of Economics and Management, Quanzhou University of Information Engineering, Quanzhou, China

\* sohailahmaduaf@yahoo.com

## Abstract

With the rapid progress of urbanization in China, the real estate industry, characterized by a long industrial chain, has become a pillar industry for economic development. Therefore, we inspect the nexus between land finance, housing prices, and economic growth. For this purpose, we use the panel data of 277 cities at the prefecture level or above in China from 2011 to 2019, and empirically examine it by using the Panel Vector Auto Regression (PVAR) model. The results show that there is a causal relationship between housing prices and economic growth. Housing prices promote economic growth in the short term and inhibit it in the long term. Both economic growth and housing prices have a significant impact on land finance. The economic growth show a significantly positive impact, while housing prices promote land finance in the short term with a long-term trend from positive to negative. This is the first study that tries to probe the relationship between urban housing prices, land finance, and economic growth by considering 277 prefecture-level and above cities in China. To promote the stable development of the regional economy, local governments need to overcome their dependence on the housing market and land finance and promote the healthy development of the housing market.

## 1. Introduction

China's urbanization level has continued to improve since the reform and opening up; the population urbanization rate increased from 17.92% in 1978 to 64.72% in 2021 [1]. The rapid advancement of urbanization has provided a powerful driving force for national economic and social development. The GDP of China has risen from 2.25 percent to 18.5 percent from 1978 to 2021 [2]. Reviewing the history of urbanization and rapid economic growth in China, it is an indisputable fact that the real estate industry drives economic growth in the fields of development, investment, production, and consumption. From 2002 to 2021, China's real estate industry's added value in GDP increased from 4.38 percent to 6.78 percent, real estate

**Data Availability Statement:** All data files are attached with Supporting Information files.

**Funding:** This work was supported by grants from the Provincial Natural Science Foundation of Hunan

(2022JJ30116), Think Tank Special Project of Social Science Fund of Hunan Province (19ZWB63), and Base Project of Social Science Fund of Hunan Province (21JD041).

**Competing interests:** The authors have declared that no competing interests exist.

development investment in GDP increased from 6.40 percent to 12.91 percent, and commercial housing sales in GDP increased from 4.96 percent to 15.91 percent [3]. Although the development process has slowed in some years, the overall trend is positive.

The vigorous development of real estate has led to the continuous rise of urban housing prices. From 2000 to 2020, the average sales price of commercial housing in China increased from 2112 yuan per m2 to 9860 yuan per m2, with an average annual growth rate of 17.46%, maintaining a rapid upward trend for a long time [4]. The impact of rising housing prices on economic growth is a double-edged sword. On the one hand, the fluctuation of housing prices is highly correlated with the fluctuation of China's GDP in time series, and the decline of housing prices is often accompanied by the decline of the GDP growth rate [5]. On the other hand, the rapid rise in house prices has exacerbated the inequality in the distribution of wealth, raising the production costs of industrial and commercial enterprises, and the resulting "de-industrialization" has weakened China's economic competitiveness [6]. The overheating of real estate investment prompted by rising house prices has also spawned a large investment bubble, and systemic financial risks are increasing. Especially since 2020, affected by the spread of the new coronavirus pneumonia epidemic and the complex and changeable international environment, the real estate industry has entered a turning point of development [7]. Many real estate-headed enterprises have experienced a "cliff-like decline," and new problems such as capital chain rupture and debt default have emerged one after another [8].

The development of the real estate industry in China is closely related to the state-owned land system of urban land [4]. After the reform of tax-sharing system in 1994, the central government collected financial power, but the local government's power did not change accordingly [9]. The mismatch of rights made the local finance department face great pressure, and even the revenue could not cover the expenditure. To make up for the financial pressure caused by the asymmetry of financial power and administrative power, local governments have to strive to seek extra-budgetary funds, and land finance with land transfer fees as the core has become the main way to make money [10]. In 2021, the transfer income from state-owned land use rights will reach 870.51 billion yuan, accounting for 78.37 percent of the local general public budget revenue [11]. Land finance has become an important driving force for economic growth and has provided major funds for industrial structure upgrading, urbanization construction, and livelihood welfare expenditures.

However, local governments' overreliance on land finance has also had a series of negative effects. The central government's assessment of local officials is a tournament model [12]. GDP and local finance are important indicators of local officials' performance assessment and promotion. On the one hand, local governments use the asset value attributes of land to promote enterprise development by curbing the rise in industrial land prices; on the other hand, they increase fiscal revenue by increasing the price of residential land transfer to meet the needs of local infrastructure construction, production, and operation [13]. Therefore, it is generally believed that land finance is one of the main reasons for the rapid rise in housing prices. Furthermore, over-reliance on land finance is likely to result in the corruption of rent-seeking officials, increased financial risk, industrial hollowing out, and so on [14].

Therefore, this paper attempts to put urban housing price, land finance, and economic growth into the same theoretical analysis framework and deeply discusses the interaction mechanism and influence effect of urban housing price, land finance, and economic growth to provide a new research perspective and theoretical basis for the Chinese government to deepen the reform of housing supply and land income distribution mechanisms and promote healthy development of the real estate market and high-quality economic development in the new development stage. suggestions.

## 2. Literature review

It is generally believed that there are "crowding in" and "crowding out" effects of housing prices on economic growth. On the one hand, rising house prices will enhance the financing capacity of enterprises, increase household wealth, and promote consumption and investment. For example, Dieppe et al. [15] found that when house prices rise by 1.5 percentage points, GDP rises by 0.4 percentage points. The static and dynamic analysis adopted by Yin and Su, [16] show that house prices and economic growth have a significant mutual promotion effect. Even household expectations of rising housing prices can increase household consumption expenditures, thereby promoting economic growth [17]. On the other hand, rising house prices have led to a decline in real income levels, reducing the consumption motivation of those who just need housing [18], pushing up the labor costs of enterprises, reducing the profitability of industrial enterprises, and thus inhibiting real economic growth [19]. According to Gelain et al. [20], the excessive rise of house prices deviating from the macroeconomic model will affect the response of companies and investors to price signals, resulting in an irrational allocation of capital and adversely affecting the economy. According to Yuan et al. [21], the effect of housing price fluctuations on economic growth exhibits significant heterogeneity of investment scale.

The problem of housing prices affecting land finance can be transformed into the relationship between housing prices and land prices [22]. Influenced by the "corn law paradox" and land rent theory, the view that house price determines land price has once again become basic common sense in the field of real estate. Based on the equilibrium theory, Deeney & O'Sullivan, [23] concluded from the perspective of demand that the reason for the high housing price is due to the high land price. Subsequently, some scholars began to use econometric models to study the relationship between house prices and land prices. For example, Joseph used Singapore's quarterly data from 1990 to 2005 as a sample, and through the Granger causality test, it was concluded that house prices were the reason for the rise in land prices and vice versa. It confirms the land rent theory proposed by Ricardo, [24]. Wu et al. [25] then used the data on house prices and land prices in 35 large and medium-sized cities in China from 2003 to 2008 for empirical analysis. The results show that land prices are determined by house prices, and high house prices are the main cause of the frequent occurrence of the "land king phenomenon." In addition, the spatial interaction between housing prices and land prices is significant. Land prices are not only affected by the demand caused by local housing prices but also by the land prices of surrounding cities [26].

In the study of the impact of land finance on economic growth, scholars generally believe that in the short term, land finance has greatly alleviated local financial pressure, provided financial support for the transformation and upgrading of industrial structures, urban infrastructure construction, and the improvement of people's livelihoods and welfare, and played a positive role in promoting economic growth [27]. The central government's tournament mode of evaluating local officials has encouraged local governments to expand land finance, drive fixed asset investment, increase enthusiasm for infrastructure construction, and directly promote economic growth [28]. The rapid advancement of the urbanization rate has also contributed to this stimulating effect [29]. In the long run, because land finance ignores the balanced development of secondary and tertiary industries and the effective allocation of resources, the economic growth model driven by fixed asset investment is not sustainable [30]. Therefore, many scholars believe that there is a Kuznets inverted U-shaped curve between land finance and economic growth—that is, land finance is conducive to economic growth in the short term, but excessive dependence on land finance will inhibit economic growth [31, 32].

On the relationship between land finance and housing prices, a large number of studies have found that land finance has a positive impact on housing prices [1]. The reform of the tax-sharing system has led to fiscal imbalances in local governments, and land finance has become an institutional factor driving house prices [33]. After controlling for other factors affecting housing prices, the more local governments rely on land finance, the faster urban house prices rise [34]. Other factors will strengthen the impact of land finance on housing prices. Wu et al. [35] put fiscal pressure, land finance, and housing prices in the same research framework and found that fiscal pressure, as an institutional factor, will indirectly affect housing prices through land finance, thus solidifying the "ratchet effect" of housing prices. The improvement of the fiscal balance level will further strengthen the role of land finance in raising housing prices [36]. In addition, due to the differences in land use efficiency, the imitation effect of local government policy formulation, and the fixed location of housing, the impact of land finance on economic growth and housing prices shows regional differences [37].

Overall, the current literature has explored the relationship between housing prices, land finance, and economic growth, but it is not yet clear how these three factors interact with each other. This article attempts to place land finance, housing prices, and economic growth within the same analytical framework, explore the interrelationships among the three, and clarify their mechanisms of action. The research contribution of this paper is to use the panel vector autoregressive model to overcome the shortcomings of the traditional time series unit root and the weakness of the coordination test. To investigate the internal correlation between urban housing prices, land finance, and economic growth, the Granger causality test, impulse response analysis, and variance decomposition are used. The research content on housing prices and land finance affecting economic growth is supplemented and improved, and the mechanism is clearer.

## 3. Theoretical analysis and research hypotheses

### 3.1 The interaction mechanism between urban housing prices and land finance

It is generally believed that land finance refers to the large-scale fiscal revenue developed by local governments in addition to the public budget revenue, which is centered on land revenue and completely controlled by local governments, including all taxes, rents, and fees obtained by the government from the process of land development, transfer and operation [38]. In terms of scale, land transfer fees are the most critical source of revenue for local governments' land finance, and since 2001, they have maintained more than 70% of direct land revenue. Under the championship model, local governments often sell industrial land at land prices to attract investment and obtain tax revenues; Selling residential land at a high price to obtain land transfer fees increases fiscal revenue [39]. Therefore, the interaction mechanism between urban housing price and land finance is essentially the mechanism of housing price and residential land price.

The land market is different from the general commodity market in that there are use planning controls. In the short term, the supply of residential land is relatively fixed, and the price of residential land depends on the level of demand, the higher the demand, the higher the land price [40]. All other things being equal, the higher the housing price, the higher the housing price, the higher the demand for residential land, which drives up land prices. In addition, since both housing and residential land have virtual asset attributes, they are both consumer goods and investment products. Rising housing prices have made it easier for real estate developers to raise capital, increasing effective demand for residential land, and financial institutions such as banks have become more confident in accepting land-collateralized financing

models, resulting in a multiplier effect [41], further pushing up land prices. For local governments, the increase in housing prices can increase the main business tax of upstream and downstream enterprises in the real estate industry and the tax revenue paid by other taxpayers, Therefore, local governments have a strong incentive to increase the price of residential land and increase land revenue by strengthening the supply and management of residential land and improving various supporting facilities.

From the perspective of enterprise cost, housing price is mainly composed of five parts: land cost, development, construction and installation cost, various taxes and fees, operating costs, and profits of real estate enterprises. Judging from the financial report data of real estate companies since 2011, land costs account for about 40% of housing prices. The price of residential land is an important factor affecting housing prices, and the increase in the price of residential land will promote the increase in housing prices. From the perspective of market supply and demand, with the acceleration of China's urbanization process, a large number of the agricultural population has turned into the urban population, the housing supply has become market-oriented, and housing demand has increased massively, housing supply has become the main factor affecting housing prices, and the supply of residential land is the key factor restricting housing supply. Under the background of the state-owned construction land, fiscal decentralization, and administrative centralization system in China, most local governments have the need and ability to endogenously generate the behavioral preference of using land transfer income to alleviate financial pressure and use it for economic construction to stimulate economic growth. Therefore, local governments maintain the supply of residential land at a low price and a high price through the strategy of controlling the total supply of residential land and transferring land transfers, which leads to a ratchet rise in housing prices.

H1. There is a significant interaction between urban housing prices and land finance.

## 3.2 The interaction mechanism between urban housing prices and economic growth

Urban housing prices have two effects on economic growth. On the one hand, urban housing prices have a positive effect on economic growth. From the perspective of composition, there is a direct correlation between housing prices and the macro-economy. From the perspective of output structure, the added value of the real estate industry is an important part of GDP, and the increase in housing sales and prices will be transmitted to the investment in the real estate industry, driving the investment and output value of the upstream and downstream related industries of real estate. From the perspective of price structure, real estate rental prices are a component of the consumer price index, and the rise in housing prices constitutes an upward pressure on nominal wages, which directly transfers labor costs to the prices of goods and services, forming inflationary pressures. Therefore, housing prices have a direct effect on economic growth. Therefore, housing prices have a direct effect on economic growth. In addition, since housing is an important form of household wealth, the increase in housing prices and the increase in nominal wealth will stimulate residents' consumption and stimulate economic growth under the effect of benign expectations. Under the effect of the collateral effect, changes in housing prices will further amplify the macroeconomic cyclical effect [42].

On the other hand, urban housing prices have a negative inhibitory effect on economic growth. The rise in the income from corporate property holdings will attract more real capital into real estate-related sectors, thus crowding out investment in other sectors of the real economy. In particular, if housing prices continue to rise rapidly, it will attract a large number of

financial resources to the real estate industry and idle internally, resulting in a bubble in real estate finance, an increase in the financing cost of the real economy, resulting in the hollowing out of the industry, resource misallocation, financial risk accumulation, and social mobility, and ultimately inhibiting economic growth [43]. For residents, housing is a consumer good, and rising housing prices indicate an expected decline in nominal and current real incomes, leading to consumption cuts. If housing is used as an investment good, the household sector will compress consumption to buy a house and repay loans, as rising housing prices create credit constraints on consumers and the need for life-cycle optimization spending.

Economic growth, in turn, affects urban housing prices. As a commodity with both investment and residential properties, the price of housing is also affected by demand and supply. From the perspective of demand, with the growth of the economy, on the one hand, the income of residents increases, which increases the demand for improved housing among urban residents. On the other hand, economic growth has promoted the urbanization rate, and a large number of rural people have been introduced to cities, thus increasing the demand for rigid housing. From the perspective of supply, economic growth has increased the supply capacity of quality housing, resulting in an increase in the supply of housing in the housing market. As a result, economic growth affects housing prices by influencing the demand and supply of housing. In addition, economic growth inevitably brings about changes in prices, which also leads to changes in the nominal price of housing. But there is often a time lag in the impact of economic growth on housing demand and supply, and therefore on housing prices.

H2. There is a significant interaction between urban housing prices and economic growth, and lag impact of economic growth on urban housing prices.

## 3.3 The interaction mechanism between land finance and economic growth

The essence of land finance is that local governments obtain land revenue through the transfer of state-owned land use rights, to make up for the shortage of fiscal revenue brought about by the transfer of financial power and administrative power after the tax sharing system. From the perspective of investment effect, local governments can use land finance to alleviate financing constraints, improve infrastructure conditions, and create conditions for increasing investment in infrastructure fixed assets, reducing taxes for enterprises and reducing land costs for enterprises, to drive investment in local industrial enterprises, promote enterprise development and achieve economic growth. From the perspective of the effect of industrial structure adjustment, land finance can use the changes in urban land structure, land use costs, infrastructure construction and land resource allocation to guide investment, promote the transition from the primary industry to the secondary industry, and the development of the secondary industry to the tertiary industry, to realize the transfer of low-end industries, the transformation of low-value-added industries to high-value-added industries, and the transformation of leading industries, to promote economic growth. From the perspective of promoting urbanization, the reform of the tax-sharing system and the championship model have prompted local governments to maximize land finance through land mortgages and urban investment bonds to promote land urbanization and population urbanization [44]. In the process of urbanization, land finance promotes the continuous expansion of land supply, provides carrying capacity for economic development, attracts rural labor to cities, increases labor production factors, accumulates human capital, and stimulates consumption expansion and upgrading, thereby it promotes the economic growth.

The key to the success of the land finance model lies in the continued high demand for urban housing in the household sector. The household sector is driven by income levels and expectations of higher housing prices. If the macroeconomic boom continues, the household sector expects future incomes to rise, and housing investment attributes strengthen, the demand for housing will be greater. However, if there is a structural adjustment in the macro-economy, the demand for housing in the household sector will decline, and the role of land finance in promoting economic growth will have a significant impact. In addition, how local governments regulate the macroeconomy through land finance is highly subjective, and the industrial upgrading that occurs is not driven by output efficiency and often ignores the balanced development of secondary and tertiary industries and the effective allocation of resources. Land finance is essentially a financial liability, and the financial resources occupied by it replace industry, resulting in an early deindustrialization effect [45], both of which have a restraining effect on economic growth. Problems such as official corruption and regional imbalances caused by the government's rent-seeking space provided by land finance will reduce the vitality of economic growth. Land finance will further widen the gap between the rich and the poor, and the crowding out effect on the real economy will hinder the sustainability of economic growth.

The impact of economic growth on land finance is also present. The core factor affecting land finance is the price of urban land, and economic growth will affect the price of land by affecting the demand and supply of land. On the one hand, economic growth will stimulate corporate investment and increase corporate demand for land. Economic growth will increase residents' demand for housing through the increase in residents' income, which will indirectly affect the demand for land by developers. On the other hand, economic growth caused by technological progress will improve the efficiency of intensive land use and increase the ability of local governments to improve construction land, thereby increasing the supply of urban construction land. However, there is often a time lag in the impact of economic growth on land demand and supply, which leads to a time lag in the impact on land prices, and therefore on land finance.

H3. There is a significant interaction between land finance and economic growth, and the impact of economic growth on land finance has a lag.

In summary, the mechanism of action between urban housing prices, land finance and economic growth is shown in Fig 1, but there are regional differences in this mechanism.

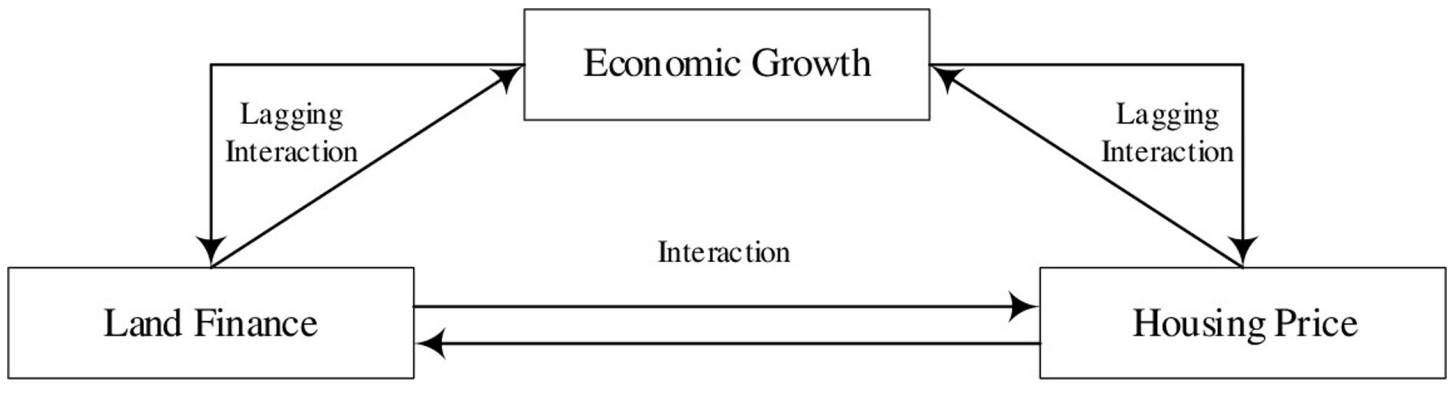

**Fig 1. Conceptual framework.**

## 4. Research design

### 4.1 Variable selection

Based on the research objectives, this paper selects five related endogenous variables based on the theory of economic growth and the research by following Hu Haisheng and Liu Hongmei [46], Han et al. [3], Cai et al. [4], Wang et al. [6], Guo & Shi [10], and Sun et al. [14]. The first one is growth in the economy: Local economic growth is generally measured by the per capita GDP scale or economic growth rate indicators, including per capita GDP at comparable prices and per capita GDP at current prices. Considering that the selected explanatory variables and control variables are mostly measured by the current price, the per capita GDP calculated by the current price is used as an indicator to measure regional economic growth. The second one is finance for land: land finance includes not only land transfer income, but also land-related real estate taxes and bank income obtained by mortgaging land. Referring to other studies [47], considering that land transfer fee income accounts for the largest proportion of land finance and that economic growth is characterized by the per capita mean value, the per capita land transfer fee is used as the proxy variable for land finance. The third one is the prices of urban housing: the average price of urban residential sales is chosen as a proxy variable for housing prices based on the practice [36]. The fourth one is residents' income level: the disposable income of urban residents in the current year is used to represent residents' income level. Lastly, the urbanization rate (UR) is the rate of population urbanization. The above non-ratio variables are processed by the natural logarithm in the regression equation. The variables involved and their metrics are shown in Table 1.

### 4.2 Model setting

As mentioned above, land finance and housing prices interact and jointly affect economic growth. Due to the lag in economic development, the current economic growth will affect housing prices and land finance in the next period. There is a correlation between land finance, housing prices, and economic growth, all of which are endogenous variables. At the same time, the research data in this paper are urban panel data. Considering this comprehensively, the Panel Vector Autoregression (PVAR) model is selected to empirically test the relationship between land finance, housing prices, and economic growth.

The PVAR model is an extension of the vector autoregression (VAR) model from planes to space (37). It was first proposed and applied by Holtz-Eakin et al. and gradually matured under the expansion of Love, Zicchino, and Lian et al. The PVAR model retains the advantage that the VAR model can regard each variable as an endogenous variable to influence all lag variables and introduces variables that consider time effects and individual effects. It combines the three-dimensional variable characteristics of the panel data model and the VAR model. The advantages of multiple interactive variables in dynamic simultaneous equations are that

**Table 1. Variables declarations.**

| Variable name | Variable code | Definition and measurement |
|---|---|---|
| economic growth | *ecog* | The per capita GDP calculated by the current price is logarithmically processed. |
| land finance | *landfin* | Per capita land transfer fee for logarithmic combing |
| Housing Prices | *prihou* | Logarithmic Processing of Urban Housing Sales Average Price |
| Income level of residents | *incomeper* | Logarithmic Processing of Urban Residents' Disposable Income in the Year |
| Urbanization rate | *urbanrate* | Urban population / total population |

they not only effectively avoid the endogeneity problem of model variables but also reduce the requirement for data length, can truly describe the relationship between variables, and can analyze the impact of a variable on other variables. Therefore, this paper constructs a PVAR model to describe the dynamic relationship between China's land finance and urban housing prices, economic growth, and the selected control variables. The following is the specific function form:

$$y_{it} = \alpha_1 + \sum_{j=1}^{p} \beta_j y_{i,t-j} + u_i + t + \varepsilon_{it} \tag{1}$$

Expand the equation.

$$y_{it} = \begin{bmatrix} ecog_{it} \\ landfin_{it} \\ prihou_{it} \\ incomeper_{it} \\ urbanrate_{it} \end{bmatrix} \tag{2}$$

$$y_{i,t-j} = \begin{bmatrix} ecog_{i,t-j} \\ landfin_{i,t-j} \\ prihou_{i,t-j} \\ incomeper_{i,t-j} \\ urbanrate_{i,t-j} \end{bmatrix} \tag{3}$$

$$\beta_j = \begin{bmatrix} \beta_{11}^{(j)} & \beta_{12}^{(j)} & \beta_{13}^{(j)} & \beta_{14}^{(j)} & \beta_{15}^{(j)} \\ \beta_{21}^{(j)} & \beta_{22}^{(j)} & \beta_{23}^{(j)} & \beta_{24}^{(j)} & \beta_{25}^{(j)} \\ \beta_{31}^{(j)} & \beta_{32}^{(j)} & \beta_{33}^{(j)} & \beta_{34}^{(j)} & \beta_{35}^{(j)} \\ \beta_{41}^{(j)} & \beta_{42}^{(j)} & \beta_{43}^{(j)} & \beta_{44}^{(j)} & \beta_{45}^{(j)} \\ \beta_{51}^{(j)} & \beta_{52}^{(j)} & \beta_{53}^{(j)} & \beta_{54}^{(j)} & \beta_{55}^{(j)} \end{bmatrix} \tag{4}$$

The equation, $ecog_{it}$、 $landfin_{it}$、 $prihou_{it}$、 $incomeper_{it}$、 $urbanrate_{it}$ represents five endogenous variables: economic growth, land finance, housing prices, income levels and urbanization rate; $i$ is the number of cities; $\beta_j$ is the regression coefficient matrix of the model variable; $j$ is a vector of endogenous variables for $j$ cities in the year $i$; $y_{i,t-j}$ is all endogenous variables of lag $j$; $\alpha_1$ is an intercept term vector; $j$ is the lag order; $\mu_i$ for individual effects; $t$ for time trends; $\varepsilon_{it}$ is a random disturbance term.

The Stata software is used in this paper to control the individual fixed effects using the Helmert transform and then the lag term of each variable as their respective instrumental variables. To avoid the estimation bias caused by the heteroscedasticity or autocorrelation of the disturbance term, the Stata panel self-vector regression program PVAR2, written by Lian Yujun to improve Love and Zicchino, is used. The more effective panel generalized moment estimation (GMM) method is used to estimate the model parameters, and the short-term interaction coefficients of economic growth, land finance, housing price, income level, and urbanization rate are obtained. At the same time, the impulse response function analysis is

used to observe the contribution of a variable, and the Granger causality test is used for each variable's causal analysis.

## 4.3 Data description

As of the end of 2020, the National Bureau of Statistics had identified 297 cities at the prefecture level and above. Based on the availability of data, this paper selects 277 prefecture-level and above cities with relatively perfect land and real estate markets as the research objects, including 34 cities in the eastern region, 86 cities in the central region, 79 cities in the western region, and 78 cities in the northeast region, accounting for 12.27%, 30.05%, and 28.52gion, 79 cities in the western region, and 78 cities in the northeast region, accounting for 12.27%, 30.05%, 28.52%, and 28.16ies in the western region, and 78 cities in the northeast region, accounting for 12.27%, 30.05%, 28.52%, and 28.16%, respectively. Urban housing prices come from the national urban housing sales average price data monitored by the Xitai database. Due to the late construction of the domestic urban housing price monitoring database and the lack of data before 2010, the selected research period is 2011–2019. The land transfer fee data are from the "China Land and Resources Statistical Yearbook" and the "China Wind Database." Since the "China Land and Resources Statistical Yearbook" was not updated after 2018, there are many missing data points in the wind database, and there are some differences in the statistical caliber of the two databases. Therefore, the data on land transfer fees in 2018 and 2019 are based on the data of the ' China Land and Resources Statistical Yearbook ' from 2015 to 2017. The AR model is used to estimate the relative coefficient, and then the wind database is fine-tuned. The remaining data are from the "China Urban Statistical Yearbook." The extreme values are winsorized, and some missing data are supplemented by interpolation. The descriptive statistics of the variables in this study are shown in Table 2.

## 5. Empirical analysis

### 5.1 Stability test

Before estimating the model, it is necessary to test the stationarity of each variable's data to avoid spurious regression. The unit root test of panel data mainly includes the LLC test, the Breiyung test, and the HT test. To ensure the reliability of the test results, this paper selects the LLC test and the HT test. The results are shown in Table 3. All variables have passed the stationarity test significantly and meet the I (0) stationarity requirements. They can be identified as stationary sequences, and there is no pseudo-regression problem.

 **5.1.1 Optimal lag order estimation.** Before establishing the regression model, it is necessary to select the optimal lag order. If the lag order is too large, it will reduce the degree of freedom and affect the effectiveness of model parameter estimation. If the lag order is too small, it will lead to a serious loss of sample data. Therefore, this paper determines the optimal lag order of the model according to the Akaike information (AIC) criterion, the Schwartz information (SIC) criterion, and the Hannan-Quine information (HQIC) criterion. The test results

**Table 2. Descriptive statistical results of variables.**

| variable | number of samples | Average value | standard deviation | Minimum value | Maximum value |
|---|---|---|---|---|---|
| ecog | 2493 | 10.715 | .584 | 8.773 | 13.056 |
| landfin | 2493 | 7.379 | 1.077 | 2.816 | 10.51 |
| prihou | 2493 | 8.66 | .483 | 7.582 | 11.073 |
| incomeper | 2493 | 10.213 | .296 | 9.348 | 11.21 |
| urbanrate | 2493 | 54.283 | 14.802 | .355 | 118.84 |

**Table 3. Unit root test results.**

| Variable | HT test | result | LLC test | result |
|---|---|---|---|---|
| *ecog* | -0.1504*** | stationary | -35.0982*** | stationary |
| *landfin* | 0.0361*** | stationary | -67.6097*** | stationary |
| *prihou* | 0.1284*** | stationary | -24.1254*** | stationary |
| *incomeper* | 0.0332*** | stationary | -1.2e+02*** | stationary |
| *unbanrate* | 0.1134*** | stationary | -32.5827*** | stationary |

Note: *, * * and * * * are significantly correlated at 10%, 5% and 1% levels, respectively.

are shown in Table 4. When the lag is 1, AIC, SIC, and HQIC all have the minimum values. According to the AIC, SIC, and HQIC minimization criteria, the lag order is determined to be 1.

**5.1.2 Model GMM estimation.** According to the AIC, SIC and HQIC criteria, the PVAR model has better estimation results when the lag order is 1. Therefore, according to Eq (1), a PVAR model with one-order lag is established:

$$y_{it} = \alpha_1 + \sum_{j=1}^{p} \beta_j y_{i,t-1} + u_i + t + \varepsilon_{it} \tag{5}$$

The GMM estimation of Eq (5) is shown in Table 5.

It can be seen from Table 5 that when economic growth is taken as the explained variable, the dynamic effects of economic growth, housing price, and residents' income level lagging one order on economic growth are significantly positive at the level of 1%, indicating that China's economic growth in the early stage has a significant role in promoting economic growth in the later stage, and the rise of housing price and residents' income level has significantly promoted economic growth. Taking land finance as the explained variable, the dynamic effects of economic growth, land finance, housing price, and residents' income level on land finance are significantly positive at the levels of 1 percent and 5 percent, respectively, indicate that China's land finance has formed a positive feedback in the time series and the increase in economic growth, housing price, and residents' income level has significantly promoted land finance. Taking housing price as the explained variable, the dynamic effects of economic growth, housing price, and residents' income level on housing price are significantly positive at the levels of 5 percent and 1 percent, respectively, indicating that with the advancement of time, China's housing price has also formed a ratchet effect, and the increase of economic growth and residents' income level has significantly promoted the rise of housing price.

**5.1.3 Impulse response function.** Because the PVAR model is dynamic, the interaction between variables is more complex, and it is difficult to accurately determine the impact of a change in one variable on other variables. The impulse response function (IRF) of PVAR refers

**Table 4. Optimal lag order selection.**

| Lag order | AIC | SIC | HQIC |
|---|---|---|---|
| 1 | 1.0881* | 5.13844* | 2.57763* |
| 2 | 1.26968 | 5.94575 | 3.00275 |
| 3 | 1.22302 | 6.73988 | 3.28645 |
| 4 | 1.37295 | 8.08804 | 3.91242 |
| 5 | 1.96682 | 10.5483 | 5.25749 |

**Table 5. GMM estimation results.**

| 变量 | ecog | landfin | prihou | incomeper | unbanrate |
|---|---|---|---|---|---|
| ecog L1 | 0.506*** (10.770) | 0.266*** (2.800) | 0.045** (2.300) | -0.001 (-0.140) | -0.123 (-0.170) |
| landfin L1 | -0.020 (-1.330) | 0.399*** (10.060) | -0.003 (-0.350) | -0.006** (-2.150) | -0.075 (-0.450) |
| prihou L1 | 0.327*** (-3.780) | 0.848*** (3.830) | 0.528*** (8.450) | -0.002 (-0.120) | -1.033 (-0.820) |
| incomeper L1 | 0.412*** (3.540) | 0.700** (-2.520) | 0.315*** (3.930) | 0.739*** (33.660) | 1.195 (1.450) |
| unbanrate L1 | 0.005 (1.020) | 0.016 (1.540) | -0.003 (-1.210) | 0.002* (1.830) | 0.776*** (11.920) |

Note: *, * * and * * * are significantly correlated at 10%, 5% and 1% levels, respectively.

to the impact of a variable in the model on each variable in the system when other variables are controlled unchanged. With the help of the impulse response function, we can intuitively reflect the dynamic relationship between variables and effectively predict the future development trend. As a result, the impulse response function (IRF) analysis of the PVAR model is performed further to illustrate the change trajectory of the interaction between land finance, housing price, and economic growth in China, as well as to describe the time lag relationship between the interaction between variables and the influence of judgment variables. In this study, 200 Monte-Carlo simulations were carried out, and the time was set to 0–10. Figs 2–4 show the impulse-response relationship between land finance, housing prices, and economic growth in China. The intermediate curve indicates the amplitude of the impact on the variable; the upper and lower sides of the curve indicate the boundary of the 95 percent confidence interval; the horizontal axis indicates the number of response periods; and the vertical axis indicates the positive and negative response and the degree of strength.

From Figs 1–3, it can be seen that when economic growth, land finance, and housing prices are impacted, the impact on themselves is a positive response that reaches a maximum in the current period and then begins to gradually decline. Among them, economic growth and land finance reach a stable value around the fourth period, while housing prices reach a stable value around the eighth period. It shows that economic growth, land finance, and housing prices have their own strengthening mechanisms, but the strengthening mechanisms gradually weaken over time.

The response of economic growth to the impact of land finance and housing prices is negative, and both show a gradual decline at first and begin to rise gradually after the second period until the tenth period tends to be stable. It shows that in the long run, land finance and housing prices have no significant positive effect on economic growth. The response of land finance and housing prices to the impact of economic growth is positive, and both show the

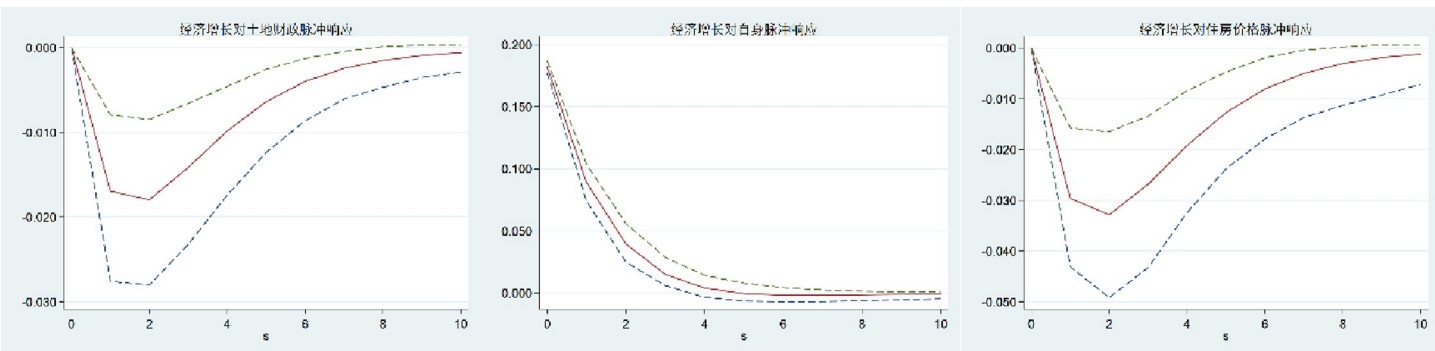

**Fig 2. Impulse response to economic growth.**

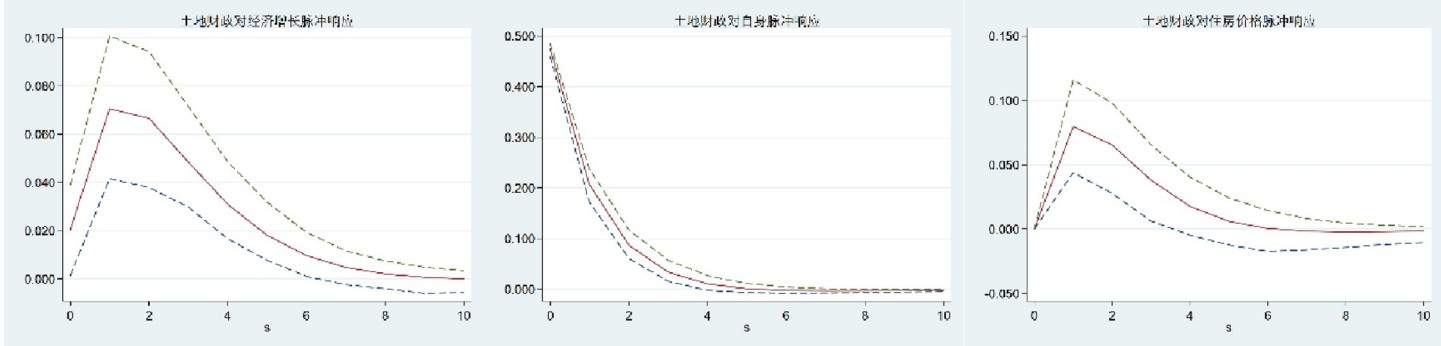

**Fig 3. Impulse response to land finance.**

characteristics of gradually rising and then gradually declining, and both tend to be stable from the 8th period on. It shows that economic growth has a positive effect on land finance and housing prices, but the trend of this promotion is inverted U-shaped. In the long run, the promotion effect is not obvious.

The response of land finance to the impact of housing prices and the response of housing prices to land finance go from positive to negative, then tend to be stable. The response of land finance to the impact of housing prices is to gradually increase and then decrease, while the response of housing prices to the impact of land finance is to decrease significantly and then gradually increase until it tends to be stable. It shows that the interaction between land finance and housing prices is negative in the long run.

**5.1.4 Variance decomposition.** The impulse response function of PVAR can reflect the dynamic influence between variables, and the variance decomposition can decompose the variance of variables into each disturbance term and analyze the contribution of each unit shock to the error variation coefficient of the predicted endogenous variables. This paper employs the variance decomposition method of PVAR and simulate 200 times to accurately measure the variance contribution rate of China's economic growth, land finance, housing prices, residents' income level, and urbanization rate. The number of periods is 20, and the contribution of each shock of each variable to a certain variable is obtained. The results are shown in Table 6.

It can be seen from Table 6 that, for economic growth, the contribution of land finance and housing prices has increased over time, from 0.007 to 0.020 and from 0.021 to 0.066, respectively. Compared with land finance, the contribution of housing prices and the increase are

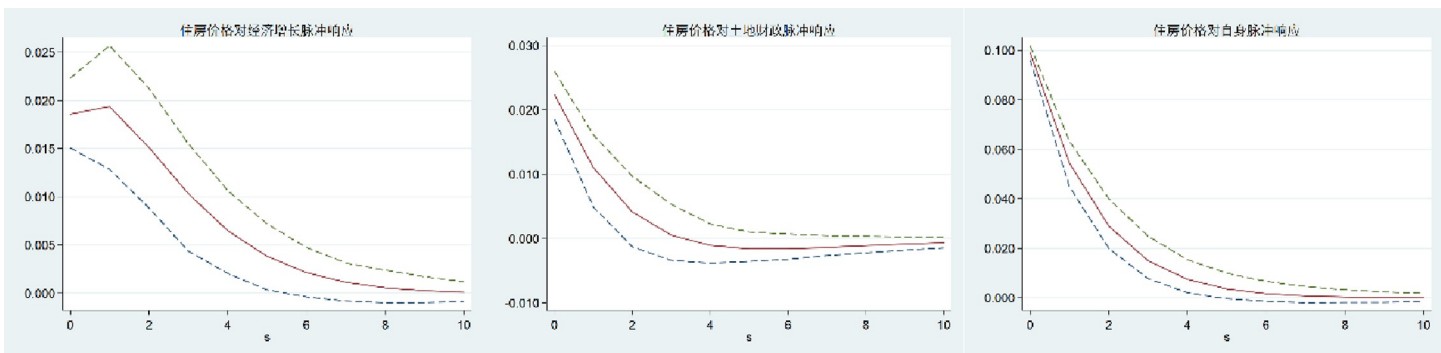

**Fig 4. Impulse response to housing price.**

**Table 6. Variance decomposition results.**

| response variable | number of periods | Impact variables | | | | |
|---|---|---|---|---|---|---|
| | | *ecog* | *landfin* | *prihou* | *incomeper* | *unbanrate* |
| *ecog* | 2 | 0.963 | 0.007 | 0.021 | 0.008 | 0.002 |
| | 5 | 0.880 | 0.019 | 0.062 | 0.021 | 0.018 |
| | 10 | 0.863 | 0.020 | 0.066 | 0.022 | 0.030 |
| | 15 | 0.862 | 0.020 | 0.066 | 0.021 | 0.031 |
| | 20 | 0.862 | 0.020 | 0.066 | 0.021 | 0.031 |
| *landfin* | 2 | 0.019 | 0.952 | 0.023 | 0.002 | 0.004 |
| | 5 | 0.043 | 0.903 | 0.040 | 0.003 | 0.011 |
| | 10 | 0.044 | 0.897 | 0.040 | 0.004 | 0.014 |
| | 15 | 0.044 | 0.897 | 0.040 | 0.004 | 0.015 |
| | 20 | 0.044 | 0.897 | 0.040 | 0.004 | 0.015 |
| *prihou* | 2 | 0.05 | 0.044 | 0.893 | 0.011 | 0.002 |
| | 5 | 0.066 | 0.039 | 0.835 | 0.055 | 0.007 |
| | 10 | 0.066 | 0.038 | 0.817 | 0.072 | 0.007 |
| | 15 | 0.065 | 0.038 | 0.816 | 0.073 | 0.007 |
| | 20 | 0.065 | 0.038 | 0.816 | 0.073 | 0.007 |
| *incomeper* | 2 | 0.02 | 0.002 | 0.024 | 0.95 | 0.004 |
| | 5 | 0.017 | 0.008 | 0.02 | 0.931 | 0.024 |
| | 10 | 0.015 | 0.010 | 0.018 | 0.913 | 0.043 |
| | 15 | 0.015 | 0.011 | 0.018 | 0.909 | 0.046 |
| | 20 | 0.015 | 0.011 | 0.018 | 0.909 | 0.047 |
| *unbanrate* | 2 | 0.004 | 0.004 | 0.001 | 0.011 | 0.98 |
| | 5 | 0.003 | 0.007 | 0.005 | 0.014 | 0.971 |
| | 10 | 0.003 | 0.008 | 0.007 | 0.014 | 0.969 |
| | 15 | 0.003 | 0.008 | 0.007 | 0.014 | 0.968 |
| | 20 | 0.003 | 0.008 | 0.007 | 0.014 | 0.968 |

greater, indicating that compared with land finance, China's economic growth is more dependent on the commercial housing market. For land finance, the contribution of economic growth and housing prices is also gradually increasing, and the final stable value is stable at about 0.4, indicating that the contribution of economic growth and housing prices to land finance is not much different, probably because land finance is more relevant to the land policies of cities. For housing prices, the contribution of economic growth is gradually increasing while the contribution of land finance is gradually decreasing, and based on the stable value, the contribution of economic growth to housing prices is greater than that of land finance. In the long run, the activity of the housing market ultimately depends on the economic fundamentals.

From the overall situation, China's land finance, housing prices, economic growth, residents' income level, and urbanization rate contribute much more to its own than other variables, indicating that these variables have their strengthening mechanisms. The variance decomposition results of the 15th forecast period and the 20th forecast period are very similar. The contribution rate of economic growth to land finance is stable at 4.4 percent, and the contribution rate to housing prices is stable at 6.5 percent. The contribution rates of land finance and housing prices to economic growth are stable at 2.0% and 6.6vel, and the urbanization rate contributes much more on its own than other variables, indicating that these variables have their strengthening mechanisms. The variance decomposition results of the 15th forecast

period and the 20th forecast period are very similar. The contribution rate of economic growth to land finance is stable at 4.4 percent, and the contribution rate to housing prices is stable at 6.5 percent. The contribution rates of land finance and housing prices to economic growth are stable at 2.0% and 6.6%, respectively. It demonstrates that the dynamic relationship between land finance, housing prices, and economic growth in China is balanced after the 15-month forecast period, and the empirical results of this paper are proven to be robust.

**5.1.5 Granger causality test.** This paper employs the Granger causality test to analyze the causal relationship between variables and distinguishes what is the cause and what is the result between variables to further clarify the causal relationship between China's land finance, housing prices, economic growth, income level, and urbanization rate, as well as the direction of short-term dynamic effects between variables. The test results are shown in Table 7.

It can be seen from Table 7 that housing price is the Granger cause of economic growth, and economic growth is also the Granger cause of housing price, indicating that there is a two-way Granger causality between China's economic growth and housing price. In addition, there is a one-way Granger causality between economic growth and land finance, and economic growth is the Granger cause of land finance; there is a one-way Granger causality between land finance and housing prices, and housing prices are the Granger cause of land finance.

This article believes that housing prices are the main Granger cause of economic growth from two aspects. On the one hand, the increasing space for urbanization rate in China and the government's repeated policy orientation of relaxing regulation of the housing market to

**Table 7. Granger causality test results.**

| Variable being explained | explanatory variables | Chi2 | df | Prob |
|---|---|---|---|---|
| ecog | landfin | 1.7759 | 1 | 0.183 |
| | prihou | 14.254 | 1 | 0.000*** |
| | incomeper | 12.534 | 1 | 0.000*** |
| | unbanrate | 1.048 | 1 | 0.306 |
| | ALL | 43.269 | 4 | 0.000*** |
| landfin | ecog | 7.8181 | 1 | 0.005*** |
| | prihou | 14.655 | 1 | 0.000*** |
| | incomeper | 6.3473 | 1 | 0.012** |
| | unbanrate | 2.3653 | 1 | 0.124 |
| | ALL | 94.406 | 4 | 0.000*** |
| prihou | ecog | 5.3121 | 1 | 0.021** |
| | landfin | 0.12557 | 1 | 0.723 |
| | incomeper | 15.464 | 1 | 0.000*** |
| | unbanrate | 1.4692 | 1 | 0.225 |
| | ALL | 33.62 | 4 | 0.000*** |
| incomper | ecog | .01905 | 1 | 0.890 |
| | landfin | 4.6266 | 1 | 0.031** |
| | prihou | .01439 | 1 | 0.905 |
| | unbanrate | 3.3438 | 1 | 0.067* |
| | All | 9.3154 | 4 | 0.054* |
| unbanrate | ecog | .02806 | 1 | 0.867 |
| | landfin | .20279 | 1 | 0.652 |
| | prihou | .67344 | 1 | 0.412 |
| | incomeper | 2.1158 | 1 | 0.146 |
| | All | 2.3455 | 4 | 0.672 |

stimulate macroeconomic growth have led to optimistic expectations of the housing market among various economic sectors in China, resulting in an amplification of the wealth effect and collateral effect multiplier of housing prices, providing sufficient capital support to drive economic growth. On the other hand, the rise in housing prices has driven an increase in government land fiscal revenue, further increasing the right of local governments to use land finance to guide advantageous industries, invest in infrastructure, and improve social welfare. This government intervention in the market has to some extent solved the problem of market failure in public goods investment in the initial stage of economic and social development, achieving the goal of government intervention to make up for market deficiencies and promote economic growth. The Granger reason for land finance is that the rise in housing prices stimulates the investment demand of real estate developers, increases the demand for urban construction land, and thus leads to an increase in land prices. The main reason for the Granger effect of economic growth on housing prices is that it leads to an increase in the nominal wealth of residents, drives their consumption expenditure, increases their demand for housing, and thus drives up housing prices.

## 5.2 Discussion

This study first investigated the connection between housing prices and land finance from the perspective of the Chinese market. The finding of our first hypothesis confirmed that housing prices and land finance are positively linked. The role of local government is also responsible for such outcomes because land revenue is the main source of revenue for local governments and is entirely under their control [38]. Furthermore, in order to draw in investment and raise taxes, local governments frequently sell industrial land at general land prices [39]. Moreover, land prices generally depend on the demand factor, and there has been an intense demand increase in the Chinese market in recent years [40]. Therefore, the government and other controlling bodies have provided land financing opportunities, which automatically enhance housing prices as well. We further probed the interactional connection amid housing prices and economic growth in China. Our second hypothesis outcome validated the above mentioned hypothesis. The real estate industry is the important factor of GDP in China and demand factor of land finance and housing prices increases the economic growth as well [42]. In particular, if home values continue to rise swiftly, a large amount of capital will be attracted to the real estate industry and sit dormant there, leading to a bubble in real estate finance. The actual economy becomes more expensive to finance as a result of this circumstance, which also eventually leads to the sector's collapse, misallocation of resources, accumulation of financial risk, obstruction of social mobility, and slowdown of economic growth [43].

Thus, over time, house prices cannot spur economic expansion. As a result of the housing price boost, a consistent rise in macroeconomic growth has raised residents' nominal wealth, reinforced their upbeat expectations, and raised demand for housing, all of which have further fueled the next round of price hikes. As a result, there is a lag in the way that economic expansion first raises home prices. We also found a relationship between land finance and the economy by taking into account the relationship between home prices and land finance.

The lack of significance of land finance on economic growth, particularly over an extended period, may be attributed to the scarcity of urban land and the stronger correlation between land finance and land policy in different cities [45]. The macroeconomy's steady expansion encourages business investment, which in turn raises land prices and increases demand for land, all of which support land finance. Thus, there is a lag in the positive relationship between economic expansion and land financing. The study adds to the body of knowledge on the links

between housing, land, and macroeconomics at the national level and offers actual data to support theories about how metropolitan area economies grow.

## 6. Conclusion and implications

### 6.1 Research findings

From the perspective of the government, housing prices, land finance, and economic growth are placed in the same analytical framework to explore the internal relationship between urban housing prices, land finance, and economic growth, which can provide a theoretical basis for the government to macro-control the housing market and formulate economic policies in different cities. Based on panel data from 2011 to 2019, this paper empirically tests the relationship between urban housing prices, land finance, and economic growth using 277 prefecture-level and above cities as the research object. The following are the findings:

1. The impact of urban housing prices and land finance on economic growth urban housing prices and land finance can positively promote economic growth in the short term, but in the long run, the positive effect is not significant. Urban housing prices are the primary cause of economic growth, but land finance is not the primary cause of economic growth. Compared with land finance, economic growth is more affected by urban housing prices. Economic growth has its strengthening mechanism, but it will gradually weaken over time, and the improvement of residents' income levels will promote economic growth.

2. The impact of land finance and economic growth on urban housing prices Land finance and economic growth will affect the city's starting price in the short term, but in the long run, land finance will have a negative impact on urban housing prices, while economic growth is not significant. Economic growth is the primary cause of urban housing prices, but land finance is not. The contribution of economic growth to urban housing prices is greater than that of land finance. The change in urban housing prices presents a ratchet effect under the influence of economic growth and land finance.

3. The impact of urban housing prices and land finance on economic growth: in the short term, urban housing prices and economic growth will promote land finance, but in the long run, urban housing prices will negatively affect land finance while economic growth is not significant. Economic growth and urban housing prices are major causes of land finance. The increase in residents' income level has also significantly promoted land finance.

### 6.2 Policy implications

Based on the above conclusions, we recommend as follows:

1. Considering the local fiscal revenue and infrastructure construction needs, one city, one policy to develop land use policy It can be seen from the empirical conclusions that land finance has a significant role in promoting urban economic growth. Therefore, we should continue to promote land marketization reform, give full play to the role of the market in the allocation of land resources, improve the efficiency of land resource use, and give full play to the positive role of land finance. Different schemes are adopted to optimize the land fiscal revenue and expenditure structures for different regional cities. the eastern region to implement the people-land policy, a reasonable determination of commercial land prices to ensure the stability of local government revenue, and a reasonable guide to land finance to

support infrastructure construction and improvement. For the central, western, and north-eastern regions, it is necessary to further improve the local government tax system, improve the sustainability of local government revenue, and coordinate local government expenditure channels.

2. Guide the healthy and stable development of the real estate market and promote the balanced development of real estate and the real economy. Because the real estate industry has many related industries and a long industrial chain, it has always been an important engine of economic growth. However, its virtual characteristics will attract a large amount of idle financial capital in the industry, resulting in "real to virtual" deviation and "virtual and real deviation," particularly during the industrial restructuring, transformation, and upgrading stages. On the one hand, the government should effectively guide reasonable expectations and demand for commercial housing purchases, boost consumers' confidence in the real estate market, fully utilize the role of market regulation, promote supply-side reform of the real estate market, promote the stable operation of housing prices, and promote the stable and healthy development of the real estate market. On the other hand, we should fundamentally attach importance to the development of the real economy, comprehensively reform the business environment from the aspects of financial support, land supply, and tax incentives, use the advantages of land resources to improve infrastructure, control the virtual economy, guide capital flows to the real economy, and promote industrial transformation and upgrading.

3. Clarify the boundary between the market and people's livelihood, and build a new model for the development of the real estate market. Strengthen the positioning of real estate as a pillar industry of the national economy, and clarify the dual housing supply system of "market + affordable". The government takes the lead in planning the construction and supply of affordable housing, implementing a dual drive of allocation-based affordable housing construction and rental-based affordable housing supply. Do a good job in the construction of public infrastructure in old residential areas and urban villages, and solve the concerns of social capital investment in urban renewal. Government regulations guide the healthy development of the real estate market. Establishing a full life cycle management mechanism for housing through special systems such as physical examinations, housing pensions, and housing insurance, guiding the market to adapt to the transformation from development-oriented to full life cycle services, and meeting the people's demand for "good housing". Gradually abolishing the system of selling properties in advance, effectively and orderly promoting the sale of existing properties, reducing the financial leverage of real estate enterprises, and reducing systemic financial risks and hidden dangers.

This article considers land finance, housing prices, and economic growth within the same analytical framework. Based on panel data from 277 cities at the prefecture level and above in China from 2011 to 2019, the panel vector autoregressive (PVAR) model is used to empirically test the causal relationship between land finance, housing prices, and economic growth in China. The research results further enrich the research on the connection between land, housing, and macroeconomics at the national scale, and provide empirical evidence for exploring the development path of urban regional economy. However, there are still some shortcomings in the research. One limitation is that due to the development level of the housing market and the difficulty in obtaining monitoring data, the research sample did not cover all prefecture-level and above cities, and the research period was only from 2011 to 2019, which is not long enough. The second is that variables such as land finance, housing prices, and economic growth all have strong locational characteristics. From a spatiotemporal perspective, spatial

factors are also key factors that need to be considered, and the model construction did not consider the introduction of spatial weights into spatial factors. These shortcomings need to be further improved in the future.

## 6.3 Limitations

However, there are still some shortcomings in the research. One limitation is that due to the development level of the housing market and the difficulty in obtaining monitoring data, the research sample did not cover all prefecture level and above cities, and the research period was only from 2011 to 2019, which is not long enough. The second is that variables such as land finance, housing prices, and economic growth all have strong locational characteristics. From a spatiotemporal perspective, spatial factors are also key factors that need to be considered, and the model construction did not consider the introduction of spatial weights into spatial factors. These shortcomings need to be further improved and perfected in the future.

## Supporting information

**S1 Data.**
(ZIP)

## Acknowledgments

The authors would like to acknowledge the support provided by the Management School, Hunan City University, Yiyang, China.

## Author Contributions

**Conceptualization:** Qian Sun.

**Data curation:** Yan Feng.

**Formal analysis:** Yan Feng.

**Methodology:** Sohail Ahmad Javeed.

**Writing – review & editing:** Yong Tang.

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
