## [Decision Letter · Decision Letter 0]

14 Feb 2024

PONE-D-24-03248The Impact of Housing Prices and Land Financing on Economic Growth: Evidence from Chinese 277 Cities at the Prefecture Level and AbovePLOS ONE

Dear Dr. Javeed,

Thank you for submitting your manuscript to PLOS ONE. After careful consideration, we feel that it has merit but does not fully meet PLOS ONE’s publication criteria as it currently stands. Therefore, we invite you to submit a revised version of the manuscript that addresses the points raised during the review process.

We look forward to receiving your revised manuscript.

Kind regards,

Shujahat Haider Hashmi, PhD Regional Economics

Academic Editor

PLOS ONE

Journal Requirements:

“This work was supported by grants from the Provincial Natural Science Foundation of Hunan (2022JJ30116), Think Tank Special Project of Social Science Fund of Hunan Province (19ZWB63), and Base Project of Social Science Fund of Hunan Province (21JD041).”

4. Please include captions for your Supporting Information files at the end of your manuscript, and update any in-text citations to match accordingly. Please see our Supporting Information 

Reviewers' comments:

Reviewer's Responses to Questions

**Comments to the Author**

1. Is the manuscript technically sound, and do the data support the conclusions?

Reviewer #1: Yes

Reviewer #2: Yes

2. Has the statistical analysis been performed appropriately and rigorously? 

Reviewer #1: Yes

Reviewer #2: Yes

3. Have the authors made all data underlying the findings in their manuscript fully available?

Reviewer #1: Yes

Reviewer #2: Yes

4. Is the manuscript presented in an intelligible fashion and written in standard English?

Reviewer #1: No

Reviewer #2: Yes

5. Review Comments to the Author

Reviewer #1: Dear authors:

This article analyzes the impact of housing prices and land financing on economic growth. The topic of the article is meaningful, but there are also some obvious errors, which are proposed for the author to modify:

1. Two words Land Finance appear in the keyword part of the first page of the PDF

2. The biggest problem of this article is that it does not conduct a theoretical analysis on the relationship between real estate, land financing and economic growth, and lacks theoretical depth. It is recommended that the author add a chapter to analyze the relationship between housing prices, land financing and economic growth before proceeding with the third part of the research design, and then develop the subsequent empirical evidence.

3. The empirical part also lacks theoretical analysis. This part is more about describing the empirical results. It lacks theoretical depth and does not explore the reasons for the results. It is recommended to supplement it.

4. In terms of language, some sentences contain grammatical errors. It is reommended that the author reorganize the language.

Kind regards.

Reviewer #2: This is an interesting and valuable academic paper. However, there are some shortcomings that need minor revision. I put forward some suggestions as follows.

1. This paper lacks elaboration on innovation, which cannot highlight how this study differs from the existing literature.

2. A figure of the research design is needed.

3. The paper lacks a discussion part, which cannot highlight the marginal contribution of the results of this study to the existing research.

4. More relevant studies especially recent ones should be included in the literature review and discussion part.

5. There are some grammar errors. Please check carefully.

6. The use of English tense and English font format in this paper is rather confusing. The normative nature of language needs to be strengthened.

7. The policy recommendations in the research conclusions are slightly insufficient. Therefore, it is suggested that the authors put forward more valuable policy suggestions according to the research results.

6. PLOS authors have the option to publish the peer review history of their article (what does this mean?). If published, this will include your full peer review and any attached files.

Reviewer #1: **Yes: **Wenfang Pu

Reviewer #2: No

---

## [Author Response · Author response to Decision Letter 0]

26 Mar 2024

Reviewer #1: Dear authors:

This article analyzes the impact of housing prices and land financing on economic growth. The topic of the article is meaningful, but there are also some obvious errors, which are proposed for the author to modify:

Response: First of all, we would like to thank reviewer for such valuable suggestions. Your suggestions really helped us to improve this paper.

Reviewer-1:

1. Two words Land Finance appear in the keyword part of the first page of the PDF

Response: Dear reviewer we have corrected the above mentioned mistake.

2. The biggest problem of this article is that it does not conduct a theoretical analysis on the relationship between real estate, land financing and economic growth, and lacks theoretical depth. It is recommended that the author add a chapter to analyze the relationship between housing prices, land financing and economic growth before proceeding with the third part of the research design, and then develop the subsequent empirical evidence.

Response: Dear reviewer, thanks for the suggestion. We have added the section as per your guidance.

3. Theoretical Analysis and Research Hypotheses

3.1 The Interaction Mechanism between Urban Housing Prices and Land Finance

It is generally believed that land finance refers to the large-scale fiscal revenue developed by local governments in addition to the public budget revenue, which is centered on land revenue and completely controlled by local governments, including all taxes, rents, and fees obtained by the government from the process of land development, transfer and operation [38]. In terms of scale, land transfer fees are the most critical source of revenue for local governments' land finance, and since 2001, they have maintained more than 70% of direct land revenue. Under the championship model, local governments often sell industrial land at land prices to attract investment and obtain tax revenues; Selling residential land at a high price to obtain land transfer fees increases fiscal revenue [39]. Therefore, the interaction mechanism between urban housing price and land finance is essentially the mechanism of housing price and residential land price.

The land market is different from the general commodity market in that there are use planning controls. In the short term, the supply of residential land is relatively fixed, and the price of residential land depends on the level of demand, the higher the demand, the higher the land price [40]. All other things being equal, the higher the housing price, the higher the housing price, the higher the demand for residential land, which drives up land prices. In addition, since both housing and residential land have virtual asset attributes, they are both consumer goods and investment products. Rising housing prices have made it easier for real estate developers to raise capital, increasing effective demand for residential land, and financial institutions such as banks have become more confident in accepting land-collateralized financing models, resulting in a multiplier effect [41], further pushing up land prices. For local governments, the increase in housing prices can increase the main business tax of upstream and downstream enterprises in the real estate industry and the tax revenue paid by other taxpayers, Therefore, local governments have a strong incentive to increase the price of residential land and increase land revenue by strengthening the supply and management of residential land and improving various supporting facilities.

From the perspective of enterprise cost, housing price is mainly composed of five parts: land cost, development, construction and installation cost, various taxes and fees, operating costs, and profits of real estate enterprises. Judging from the financial report data of real estate companies since 2011, land costs account for about 40% of housing prices. The price of residential land is an important factor affecting housing prices, and the increase in the price of residential land will promote the increase in housing prices. From the perspective of market supply and demand, with the acceleration of China's urbanization process, a large number of the agricultural population has turned into the urban population, the housing supply has become market-oriented, and housing demand has increased massively, housing supply has become the main factor affecting housing prices, and the supply of residential land is the key factor restricting housing supply. Under the background of the state-owned construction land, fiscal decentralization, and administrative centralization system in China, most local governments have the need and ability to endogenously generate the behavioral preference of using land transfer income to alleviate financial pressure and use it for economic construction to stimulate economic growth. Therefore, local governments maintain the supply of residential land at a low price and a high price through the strategy of controlling the total supply of residential land and transferring land transfers, which leads to a ratchet rise in housing prices.

H1. There is a significant interaction between urban housing prices and land finance.

3.2 The Interaction Mechanism between Urban Housing Prices and Economic Growth

Urban housing prices have two effects on economic growth. On the one hand, urban housing prices have a positive effect on economic growth. From the perspective of composition, there is a direct correlation between housing prices and the macro-economy. From the perspective of output structure, the added value of the real estate industry is an important part of GDP, and the increase in housing sales and prices will be transmitted to the investment in the real estate industry, driving the investment and output value of the upstream and downstream related industries of real estate. From the perspective of price structure, real estate rental prices are a component of the consumer price index, and the rise in housing prices constitutes an upward pressure on nominal wages, which directly transfers labor costs to the prices of goods and services, forming inflationary pressures. Therefore, housing prices have a direct effect on economic growth. Therefore, housing prices have a direct effect on economic growth. In addition, since housing is an important form of household wealth, the increase in housing prices and the increase in nominal wealth will stimulate residents' consumption and stimulate economic growth under the effect of benign expectations. Under the effect of the collateral effect, changes in housing prices will further amplify the macroeconomic cyclical effect [42].

On the other hand, urban housing prices have a negative inhibitory effect on economic growth. The rise in the income from corporate property holdings will attract more real capital into real estate-related sectors, thus crowding out investment in other sectors of the real economy. In particular, if housing prices continue to rise rapidly, it will attract a large number of financial resources to the real estate industry and idle internally, resulting in a bubble in real estate finance, an increase in the financing cost of the real economy, resulting in the hollowing out of the industry, resource misallocation, financial risk accumulation, and social mobility, and ultimately inhibiting economic growth [43]. For residents, housing is a consumer good, and rising housing prices indicate an expected decline in nominal and current real incomes, leading to consumption cuts. If housing is used as an investment good, the household sector will compress consumption to buy a house and repay loans, as rising housing prices create credit constraints on consumers and the need for life-cycle optimization spending.

Economic growth, in turn, affects urban housing prices. As a commodity with both investment and residential properties, the price of housing is also affected by demand and supply. From the perspective of demand, with the growth of the economy, on the one hand, the income of residents increases, which increases the demand for improved housing among urban residents. On the other hand, economic growth has promoted the urbanization rate, and a large number of rural people have been introduced to cities, thus increasing the demand for rigid housing. From the perspective of supply, economic growth has increased the supply capacity of quality housing, resulting in an increase in the supply of housing in the housing market. As a result, economic growth affects housing prices by influencing the demand and supply of housing. In addition, economic growth inevitably brings about changes in prices, which also leads to changes in the nominal price of housing. But there is often a time lag in the impact of economic growth on housing demand and supply, and therefore on housing prices.

H2. There is a significant interaction between urban housing prices and economic growth, and lag impact of economic growth on urban housing prices.

3.3 The Interaction Mechanism between Land Finance and Economic Growth

The essence of land finance is that local governments obtain land revenue through the transfer of state-owned land use rights, to make up for the shortage of fiscal revenue brought about by the transfer of financial power and administrative power after the tax sharing system. From the perspective of investment effect, local governments can use land finance to alleviate financing constraints, improve infrastructure conditions, and create conditions for increasing investment in infrastructure fixed assets, reducing taxes for enterprises and reducing land costs for enterprises, to drive investment in local industrial enterprises, promote enterprise development and achieve economic growth. From the perspective of the effect of industrial structure adjustment, land finance can use the changes in urban land structure, land use costs, infrastructure construction and land resource allocation to guide investment, promote the transition from the primary industry to the secondary industry, and the development of the secondary industry to the tertiary industry, to realize the transfer of low-end industries, the transformation of low-value-added industries to high-value-added industries, and the transformation of leading industries, to promote economic growth. From the perspective of promoting urbanization, the reform of the tax-sharing system and the championship model have prompted local governments to maximize land finance through land mortgages and urban investment bonds to promote land urbanization and population urbanization [44]. In the process of urbanization, land finance promotes the continuous expansion of land supply, provides carrying capacity for economic development, attracts rural labor to cities, increases labor production factors, accumulates human capital, and stimulates consumption expansion and upgrading, thereby it promotes the economic growth.

The key to the success of the land finance model lies in the continued high demand for urban housing in the household sector. The household sector is driven by income levels and expectations of higher housing prices. If the macroeconomic boom continues, the household sector expects future incomes to rise, and housing investment attributes strengthen, the demand for housing will be greater. However, if there is a structural adjustment in the macro-economy, the demand for housing in the household sector will decline, and the role of land finance in promoting economic growth will have a significant impact. In addition, how local governments regulate the macroeconomy through land finance is highly subjective, and the industrial upgrading that occurs is not driven by output efficiency and often ignores the balanced development of secondary and tertiary industries and the effective allocation of resources. Land finance is essentially a financial liability, and the financial resources occupied by it replace industry, resulting in an early deindustrialization effect [45], both of which have a restraining effect on economic growth. Problems such as official corruption and regional imbalances caused by the government's rent-seeking space provided by land finance will reduce the vitality of economic growth. Land finance will further widen the gap between the rich and the poor, and the crowding out effect on the real economy will hinder the sustainability of economic growth.

The impact of economic growth on land finance is also present. The core factor affecting land finance is the price of urban land, and economic growth will affect the price of land by affecting the demand and supply of land. On the one hand, economic growth will stimulate corporate investment and increase corporate demand for land. Economic growth will increase residents' demand for housing through the increase in residents' income, which will indirectly affect the demand for land by developers. On the other hand, economic growth caused by technological progress will improve the efficiency of intensive land use and increase the ability of local governments to improve construction land, thereby increasing the supply of urban construction land. However, there is often a time lag in the impact of economic growth on land demand and supply, which leads to a time lag in the impact on land prices, and therefore on land finance.

H3. There is a significant interaction between land finance and economic growth, and the impact of economic growth on land finance has a lag.

3. The empirical part also lacks theoretical analysis. This part is more about describing the empirical results. It lacks theoretical depth and does not explore the reasons for the results. It is recommended to supplement it.

Response: Dear reviewer, thanks for the suggestion. We have added the section as per your guidance.

This article believes that housing prices are the main Granger cause of economic growth from two aspects. On the one hand, the increasing space for urbanization rate in China and the government's repeated policy orientation of relaxing regulation of the housing market to stimulate macroeconomic growth have led to optimistic expectations of the housing market among various economic sectors in China, resulting in an amplification of the wealth effect and collateral effect multiplier of housing prices, providing sufficient capital support to drive economic growth. On the other hand, the rise in housing prices has driven an increase in government land fiscal revenue, further increasing the right of local governments to use land finance to guide advantageous industries, invest in infrastructure, and improve social welfare. This government intervention in the market has to some extent solved the problem of market failure in public goods investment in the initial stage of economic and social development, achieving the goal of government intervention to make up for market deficiencies and promote economic growth. The Granger reason for land finance is that the rise in housing prices stimulates the investment demand of real estate developers, increases the demand for urban construction land, and thus leads to an increase in land prices. The main reason for the Granger effect of economic growth on housing prices is that it leads to an increase in the nominal wealth of residents, drives their consumption expenditure, increases their demand for housing, and thus drives up housing prices.

4. In terms of language, some sentences contain grammatical errors. It is reommended that the author reorganize the language.

Response: Dear reviewer thank you for the suggestion. We have gained the professional English services to further improve the writing of this paper.

Kind regards.

At the end, we thank you again for helping us to further improve the quality of this paper.

Reviewer #2: This is an interesting and valuable academic paper. However, there are some shortcomings that need minor revision. I put forward some suggestions as follows.

Response: Dear reviewer we’re highly thankful for your time and efforts. Your suggestions really helped us to improve the quality of this paper.

1. This paper lacks elaboration on innovation, which cannot highlight how this study differs from the existing literature.

Response: Dear reviewer we have updated in the lit

---

## [Decision Letter · Decision Letter 1]

5 Apr 2024

The Impact of Housing Prices and Land Financing on Economic Growth: Evidence from Chinese 277 Cities at the Prefecture Level and Above

PONE-D-24-03248R1

Dear Dr. Javeed,

We’re pleased to inform you that your manuscript has been judged scientifically suitable for publication and will be formally accepted for publication once it meets all outstanding technical requirements.

Kind regards,

Shujahat Haider Hashmi, PhD Regional Economics

Academic Editor

PLOS ONE

Additional Editor Comments (optional):

Reviewers' comments:

Reviewer's Responses to Questions

**Comments to the Author**

1. If the authors have adequately addressed your comments raised in a previous round of review and you feel that this manuscript is now acceptable for publication, you may indicate that here to bypass the “Comments to the Author” section, enter your conflict of interest statement in the “Confidential to Editor” section, and submit your "Accept" recommendation.

Reviewer #1: All comments have been addressed

Reviewer #2: All comments have been addressed

2. Is the manuscript technically sound, and do the data support the conclusions?

Reviewer #1: Yes

Reviewer #2: Yes

3. Has the statistical analysis been performed appropriately and rigorously? 

Reviewer #1: Yes

Reviewer #2: Yes

4. Have the authors made all data underlying the findings in their manuscript fully available?

Reviewer #1: Yes

Reviewer #2: Yes

5. Is the manuscript presented in an intelligible fashion and written in standard English?

Reviewer #1: Yes

Reviewer #2: Yes

6. Review Comments to the Author

Reviewer #1: Thank you very much for your modificatio. The revised article is good and it is recommended to publish it

Reviewer #2: After revision, this paper has been highly improved. I suggest accepting this paper in its present form.

7. PLOS authors have the option to publish the peer review history of their article (what does this mean?). If published, this will include your full peer review and any attached files.

Reviewer #1: **Yes: **Wenfang Pu

Reviewer #2: No
